

# RNA N6-methyladenosine reader IGF2BP3 promotes acute myeloid leukemia progression by controlling stabilization of EPOR mRNA

Jin Fan[1], Mengqi Zhuang[2], Wei Fan[3] and Ming Hou[1]

[1] Qilu Hospital of Shandong University, Jinan, China
[2] The Fourth Hospital of Jinan, Jinan, China
[3] Department of Pharmacy and Medical Laboratory, Heze Medical College, Heze, China

## ABSTRACT

**Background:** N6-methyladenosine (m6A) methylation epigenetically regulates normal hematopoiesis and plays a role in the pathogenesis of acute myeloid leukemia (AML). However, its potential value for prognosis remains elusive.

**Methods:** Analysis of the datasets downloaded from The Cancer Genome Atlas and Genotype Tissue Expression databases revealed that the expression level of 20 regulators related to m6A RNA methylation differ between patients with AML and normal individuals. A prognostic risk model with three genes (*YTHDF3*, *IGF2BP3*, and *HNRNPA2B1*) was developed using univariate Cox regression and the least absolute shrinkage and selection operator Cox regression methods.

**Results:** This established signature demonstrated good predictive efficacy with an area under the curve of 0.892 and 0.731 in the training cohort and the validation cohort, respectively. Patients with AML and an increased level of *Insulin growth factor 2 mRNA binding protein 3 (IGF2BP3)* expression exhibited a poor prognosis. *IGF2BP3* knockdown significantly induced G0/G1 phase arrest and inhibited cell proliferation, apoptosis, and/or differentiation. Further, the JAK/STAT pathway may be involved in the regulation of EPOR expression by *IGF2BP3*-mediated m6A RNA methylation.

**Conclusion:** These findings indicate that IGF2BP3 plays a carcinogenic role in AML, implying that it can predict patient survival and could be an effective strategy for AML therapy.

## INTRODUCTION

Acute myeloid leukemia (AML) is a highly heterogeneous disease caused by a disorder of blood and bone marrow cells. In addition to genetic mutations and chromosomal variants, epigenetic abnormalities, such as histone modifications or DNA methylation, have been associated with its development (*Ye et al., 2016*). Recent studies have suggested that N6-methyladenosine (m6A) modification of RNA is involved in tumorigenesis and cancer progression.

Corresponding author
Ming Hou, qlhouming@sina.com

N6-methyladenosine methylation, an mRNA modification, is a biological process controlled by methylation transferases and demethyltransferases (*Zhang et al., 2021*). These regulators can be recognized by methylation-related reading proteins; moreover, they have various effects on mRNA, including RNA splicing, translation, decay, and other types of RNA metabolic processes (*Roundtree et al., 2017*; *Fu et al., 2014*). Furthermore, m6A mRNA methylation can affect normal hematopoiesis by preventing self-renewal of cells. It has also been linked to hematological malignancies, such as AML. For example, m6A mRNA methylation regulates tumorigenesis and tumor progression by influencing the proliferation and differentiation of normal hematopoietic and leukemic cells and mediating treatment resistance by influencing the sensitivity of leukemic cells to drugs (*Vu et al., 2017*). Furthermore, YTHDF2 can be considered a dual target for eliminating leukemia stem cells and increasing the production of normal hematopoietic stem cells, which ultimately lead to bone marrow reconstitution (*Paris et al., 2019*). FTO-targeted inhibitors significantly improve overall survival in cases of AML and breast cancer (*Li et al., 2017*). Hence, targeting or inhibiting m6A-related factors has promising therapeutic potential in AML therapy.

Although many studies have highlighted the importance of m6A mRNA methylation in AML and its potential value in prognosis, further information remains elusive. In this study, we developed a prognostic model based on three genes related to m6A mRNA methylation and validated its efficacy in a group of patients with AML. We found that insulin growth factor 2 mRNA binding protein 3 (IGF2BP3) increases the stability of methylation by recognizing and binding m6A-modified mRNA of erythropoietin receptor (EPOR), thereby activating JAK/STAT signaling pathway to promote AML progression. This indicated that it may be a promising prognostic biomarker and therapeutic target for AML.

## MATERIALS AND METHODS

### Data resources

The TCGA database (https://portal.gdc.cancer.gov/) includes datasets on gene expression and clinical resources (including age, gender, leukocyte count, and cytogenetic risk) of 151 AML patients. Patients with insufficient resources were excluded. The RNA-seq data from 70 normal individuals were collected from the Genotype Tissue Expression (GTEx) database (https://gtexportal.org/home/datasets). Based on the relevant literature, 20 m6A regulators were identified, including 10 RNA binding proteins (YTHDF1/2/3, YTHDC1/2, HNRNPA2B1, HNRNPC, and IGF2BP1/2/3), eight methyltransferases (WTAP, METTL3, METTL14, KIAA1429, RBM15, RBM15B, RBMX, and ZC3H13), and two demethylases (FTO and ALKBH5) (*Li et al., 2019*). The "sva" package was used to remove the bulk effect when combining standardized RNA-seq datasets from TCGA and GTEx databases to eliminate the influence between different datasets. The GSE37642 dataset (GPL96 platform) from the GEO database (http://www.ncbi.nlm.nih.gov/geo/) were selected as the validation cohort. Download the raw microarray data of 417 AML patients and removed the batch effects using the "limma" package.

## Construction and validation of a prognostic model

The genes associated with m6A RNA methylation were identified as prognostic indicators based on the results of univariate Cox regression. They were further validated using least absolute shrinkage and selection operator (LASSO) Cox regression based on the "glment" package. The risk scores were calculated by minimizing the regression coefficients using the following formula: Risk score = Gene1 expression $\times \beta1$ + Gene2 expression $\times \beta2$ + ... + Gene expression $\times \beta n$, where $\beta$ is the regression coefficient calculated using the multivariate Cox regression (*Zhang et al., 2018*).

The Kaplan-Meier (K-M) survival curve was used to indicate the survival rate over time, and was drawn using the "surviva" package. The *P*-value was calculated using log-rank tests. The receiver operating characteristic (ROC) curve was used to evaluate the prognostic model based on the measured risk scores. We used the "survivalROC" package to measure the area under the curve (AUC). Further, univariate and multivariate Cox regression analyses were used to identify prognosis-related factors.

Gene set enrichment analysis (GSEA) was used to identify the associated pathways in the high-risk group, and statistical significance was defined as normalize enrichment score (NES) >1, adjusted $P < 0.05$, and false discovery rate $q < 0.25$.

## Cell culture and lentivirus infection

KG-1a and THP-1 cells were purchased from the Institute of Biochemistry and Cell Biology of Chinese Academy of Sciences (Shanghai, China) and cultured in freshly prepared medium (Procell, Wuhan, China) with 10% fetal bovine serum (FBS, Gibco, Carlsbad, CA, USA) and 1% penicillin-streptomycin (Meilunbio, Dalian, China) for aseptic growth at constant temperature (37 °C and 5% $CO_2$) and passaged once every 2–3 days. Three shRNAs targeting IGF2BP3 and one scrambled RNA provided by Shanghai GenePharma Co., Ltd were cloned into a vector (Clontech, Palo Alto, CA, USA). The shRNA sequences were as follows: shIGF2BP3-1: AGGAATTGACGCTGTATAA, shIGF2BP3-2: AGTTGTAAATGTAACCTAT, shIGF2BP3-3: ATGACATTTAATTCCTGGATTA, and shNC: TTCTCCGAACGTGTCACGT. Further, puromycin (Sigma-Aldrich, St. Louis, MO, USA) selection was used to create cell lines that stably expressed shRNAs.

## Western blotting

The cells were washed twice with pre-cooled phosphate buffered saline (PBS, Boster, Pleasanton, CA, USA) and lysed with ripa (Beyotime, Shanghai, China) for 30 min; subsequently, they were centrifuged at 4 °C for 10 min at RCF: 14,800×$g$. Following this, supernatant was collected, and cells were stored at −70 °C after quantitating with a BCA kit (Beyotime, Shanghai, China). Thereafter, sodium dodecyl sulfate polyacrylamide gel electrophoresis (SDS-PAGE; Servicebio, Wuhan, China) electrophoresis was used to separate or purify interested proteins from 20 µg of total protein. After electrically transferring the proteins on the gel to the PVDF membrane (Millipore, Burlington, MA, USA), the supernatant was blocked with 5% skim milk and incubated overnight at 4 °C with primary antibodies, such as IGF2BP3 (1:1,000; Proteintech, Chicago, IL, USA), EPOR (1:2,000; Proteintech, Chicago, IL, USA), p-JAK2 (1:2,000; Proteintech, Chicago, IL, USA),

p-STAT5 (1:2,000; Proteintech, Chicago, IL, USA), C-myc (1:1,000; Proteintech, Chicago, IL, USA), Bcl-2 (1:1,000; Proteintech, Chicago, IL, USA), and GAPDH (1:5,000; Proteintech, Chicago, IL, USA). A secondary antibody labeled with horseradish peroxidase was incubated at 37 °C for 2 h before developing membranes after three washes with TBST (ServiceBio, Wuhan, China).

## Cell proliferation, apoptosis, cycle, and differentiation assay

Cell viability was determined using the Cell Counting Kit-8 (CCK-8; Beyotime, Shanghai, China). Each 96-well plate was seeded with $1 \times 10^4$ cells, and each sample had ≥3 replicates. Cell viability was measured at several time points after transfection, *i.e.*, at 0, 12, 24, and 48 h. Notably, 10 μL of CCK-8 was incubated in each well at 37 °C for 2 h before measuring absorbance at a wavelength of 490 nm using a microplate reader.

Flow cytometry was used to detect apoptotic cells using the Annexin V-PE/7-AAD (7-Aminoactinomycin D) Apoptosis Detection Kit (Meilunbio, Dalian, China). The cells were digested with trypsin/EDTA (Gibco, Carlsbad, CA, USA); subsequently, they were centrifuged, suspended, and incubated for 15 min in 100 μL of solution containing 5 μL of Annexin V-PE and 7-AAD each; they were then detected using flow cytometry (Agilent, Santa Clara, CA, USA). The data were analyzed using CellQuest software (Becton Dickinson, Franklin Lakes, NJ, USA) to plot a two-color graph.

Cell Cycle and Apoptosis Analysis Kit (Meilunbio, Dalian, China) was used to detect cell cycle. The cells were fixed overnight in 70% ethanol, resuspended, and incubated for 20 min in 100 μL of PBS containing 0.01% bovine serum albumin (BSA), 100 mg/mL of RNAase, and 2 μL of propidium iodide (PI). Further, flow cytometry was performed using a novocyte instrument (Agilent, Santa Clara, CA, USA).

Subsequently, phorbol 12-myristate 13-acetate (PMA, Sigma, Springfield, MO, USA)-treated cells (200 ng/mL, 48 h) were lysed, and RNA was extracted. The expression of cell differentiation-related markers CD11b and CD14 was examined using real-time quantitative PCR (qPCR).

## Real-time quantitative PCR

The total RNA of KG-1a or THP-1 cells was extracted by TRIzol Reagent (Cwbio, Beijing, China). Further, the concentration of RNA was determined using an ultraviolet spectrophotometer. Complementary DNA (cDNA) was prepared using the rapid reverse transcription kit (Accurate Biology, Hunan, China). Real-time quantitative PCR was performed using the Kapa SYBR Green Pro Taq HS qPCR Kit (Accurate Biology, Hunan, China). The reaction system of 20 μL was set at 95 °C for 20 s, 95 °C for 3 s, 60 °C for 20 s, 72 °C for 20 s, and 40 cycles. The gene-specific qPCR primers were synthesized using Sangon Biotech (Shanghai, China). GAPDH was selected as an internal reference for calibration and standardization. The primer sequences were as follows: IGF2BP3-F: 5′-AGGCTCAGTTCAAGGCTCAG-3′, IGF2BP3-R: 5′-TGGTCATTCTCATCAGGTGTCT-3′, CD11b-F: 5′-TTGGTGGCTTCCTTGTGGTT-3′, CD11b-R: 5′-GTAGTCGCACTGGTAGAGGC-3′, CD14-F: 5′-AAGCACTTCCAGAGCCTGTC-3′,

CD14-R: 5′-TCGTCCAGCTCACAAGGTTC-3′, EPOR-F: 5′-TTCTGGTGTTCGCTGCCTA-3′, EPOR-R: 5′-ACGTCATGGGTGTCTCAGG-3′, and GAPDH-F: 5′-AGAAGGCTGGGGCTCATTTG-3′, GAPDH-R: 5′-AGGGGCCATCCACAGTCTTC-3′.

## Methylated RNA immunoprecipitation qPCR

Total RNA was extracted using TRIzol Reagent (Cwbio, Beijing, China). Further, 200 μg of RNA was fragmented into fragments sized 200 nt by adding a fragmentation reagent, and the fragmented RNA was divided into two parts. Subsequently, immunomagnetic beads with pre-mixed m6A antibody (1:100; German Synaptic Systems, Göttingen, Germany) were added to enrich the mRNA fragment containing m6A methylation. The remaining samples were used as input controls. After 2 h of incubation at 4 °C, the samples were washed five times with high and low salt detergents. Notably, proteinase K digestive solution digests magnetic beads, releases enriched RNAs, and purifies RNA using RNA Purification Kit (Accurate Biology, Hunan, China). cDNA was synthesized using a rapid reverse transcription kit and used for subsequent quantitative PCR.

## RNA stability assay

Cell lines were stably expressing shIGF2BP3, and scrambled RNA samples were cultured in six-well plates. Cells were collected at 0, 2, and 4 h after treatment with 5 μg/mL of actinomycin D (APExBIO, Houston, TX, USA). Total RNA was extracted, and the mRNA level of EPOR was determined using qPCR.

## Bioinformatic and statistical analyses

Overall, 20 genes that were related to m6A RNA methylation and differentially expressed in patients with AML and normal individuals were identified using the R software (Version 4.1.0) "limma" packages. The "pheatmap" and "vioplot" packages were used to create heat map and violin map to visualize differences in gene expression. Pearson's correlation coefficient was used to determine the correlation between these genes. Patients with AML were classified into two groups using the package "ConsensusClusterPlus". Principal component analysis (PCA) was used to visualize the clusters in order to exclude deviations. Furthermore, the K-M survival curve was used to evaluate overall survival. We used the "pheatmap" package to visualize the expression differences of 20 regulators between the two subgroups and further compared the clinical characteristics of the two subgroups.

Data were analyzed using GraphPad Prism 8.0 and SPSS (Version 22.0; IBM Corp, Armonk, NY, USA). Chi-square test was used to determine the statistical significance of the correlation between *IGF2BP3* and clinical features. Student's *t*-test was used to compare the two groups (AML and control). Survival data were analyzed using univariate and multivariate Cox regression analyses. Overall survival was demonstrated using K-M survival curves, and statistical significance was determined using the log-rank test. $P$-values of $<0.05$ were used to define statistically significant results.

## RESULTS

### Expression of genes related to m6A mRNA methylation in the AML and control groups

Regarding the expression of 20 genes related to m6A RNA methylation, we found that the expression significantly differs between the AML and control groups (Fig. 1A); these genes included five downregulated genes (*METTL14*, *WTAP*, *HNRNPC*, *KIAA1429*, *FTO*, and *WTAP*) and 15 upregulated genes (*IGF2BP1*, *IGF2BP3*, *YTHDF1*, *YTHDF2*, *YTHDF3*, *ALKBH5*, *METTL3*, *RBM15*, *RBM15B*, *YTHDC1*, *YTHDC2*, *IGF2BP2*, *ZC3H13*, *HNRNPA2B1*, and *RBMX*) (Fig. 1B). Correlation analysis (Fig. 1C) of 20 genes regulating m6A RNA methylation indicated that 0.88 was the highest correlation coefficient between the two genes, *i.e.*, between *RBMX* and *HNRNPA2B1*, *RBMX* and *YTHDF2*, *RBMX* and *ZC3H13*, and *YTHDF2* and *ZC3H13*.

### Identification of the two clusters of AML samples

AML samples were further divided into two groups based on the transcriptional gene expression (Fig. 2A), and the results of principal component analysis revealed no significant differences between the two subgroups (Fig. 2B). The analysis of overall survival revealed that the survival time of subgroup 2 was higher than that of subgroup 1 (Fig. 2C). However, no significant correlation was found between prognosis and patients' clinical characteristics (Fig. 2D). This could be attributed to the small sample size or the sensitivity of the clustering algorithm to the data.

### Establishment of a prognostic model comprising three genes related to m6A mRNA methylation

Univariate (Fig. 3A) and LASSO Cox regression (Figs. 3B and 3C) analyses were performed on the expression profile data to identify potential prognostic factors from the genes related to m6A RNA methylation. Three genes (*IGF2BP3*, *HNRNPA2B1*, and *YTHDF3*) were selected as AML prognostic model. Among them, *IGF2BP3* with HR >1 was considered a risky gene, whereas *HNRNPA2B1* and *YTHDF3* with HR <1 were considered protective genes. These three genes had coefficients of 2.538092, −0.020233, and −0.068759, respectively. Subsequently, we evaluated each patient's risk scores in TCGA training cohort and the GSE37642 validation cohort. Further, all patients were classified into two groups (high- and low-risk groups) using the median risk score as a cut-off. The K-M survival analysis revealed that patients in the high-risk group had significantly shorter survival times ($P = 1.41 \times 10^{-5}$, Fig. 3D and $P = 5.034 \times 10^{-8}$, Fig. 3F). We calculated the area under the curve (AUC) again to validate the efficacy of our prognostic model. The data revealed that the calculated AUC was 0.892 (Fig. 3E) and 0.731 (Fig. 3G) respectively, which indicated that the three-gene signature is a favorable prognosis model. Notably, risk score and survival status indicated that the survival time of patients with high-risk scores was shortened (Figs. 3H and 3I).

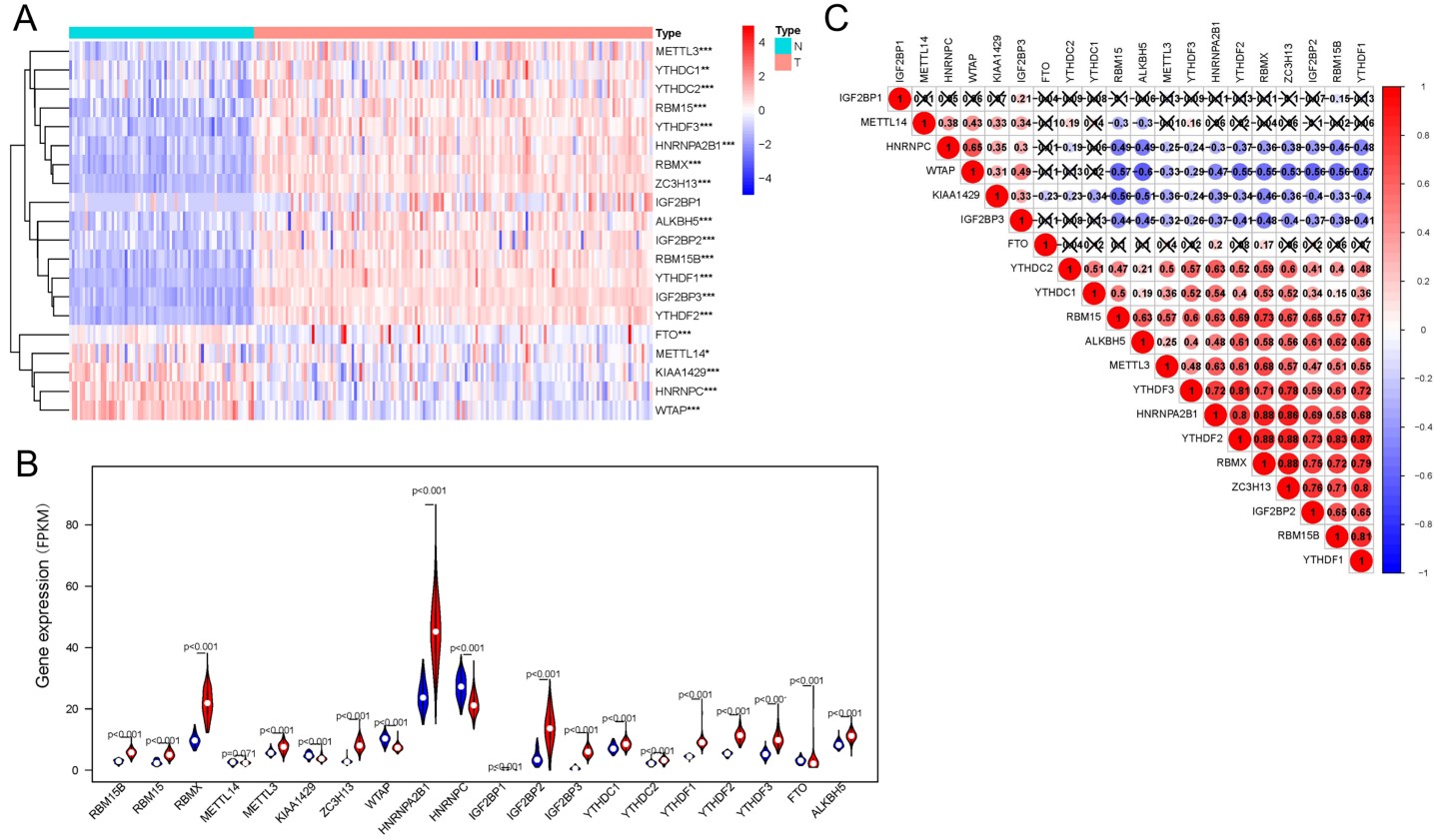

**Figure 1 Expression of and correlation among m6A methylation regulators.** (A) Heatmaps of m6A-related genes expressed in patients with AML and in normal individuals. (B) Vioplot shows the expression of m6A-related genes in patients with AML. Blue represents the control group, whereas red represents the AML group. (C) Correlation matrix of interaction among m6A methylation regulators. Correlation coefficients are plotted by indicating negative correlations in blue and positive correlations in red. *$P < 0.05$, **$P < 0.01$, ***$P < 0.001$.

### Determination of the signature comprising m6A methylation-related genes as an independent prognostic factor

We performed the Cox regression analysis again to confirm whether the established signature was an independent prognostic factor. The results suggested that age, risk score, and cytogenetics risk were independent prognostic factors in AML (Table 1).

The pathway involved in tumor proliferation was significantly enriched in the high-risk group. Other pathways were as follows: chemokine signaling pathway, VEGF signaling pathway, pathway in the activity of hematopoietic cell lineage or AML, MAPK signaling pathway, cell adhesion molecule-associated pathways, NOTCH signaling pathway, gap junction, apoptosis, and TOLL pathways (Fig. 4A).

### Highly expressed IGF2BP3 in patients with AML as an independent prognostic factor

In order to understand the impact of every gene in the prognosis model consisted of IGF2BP3, HNRNPA2B1, and YTHDF3 on prognosis, the OS of patients with high expression were compared with those of patients with low expression respectively.

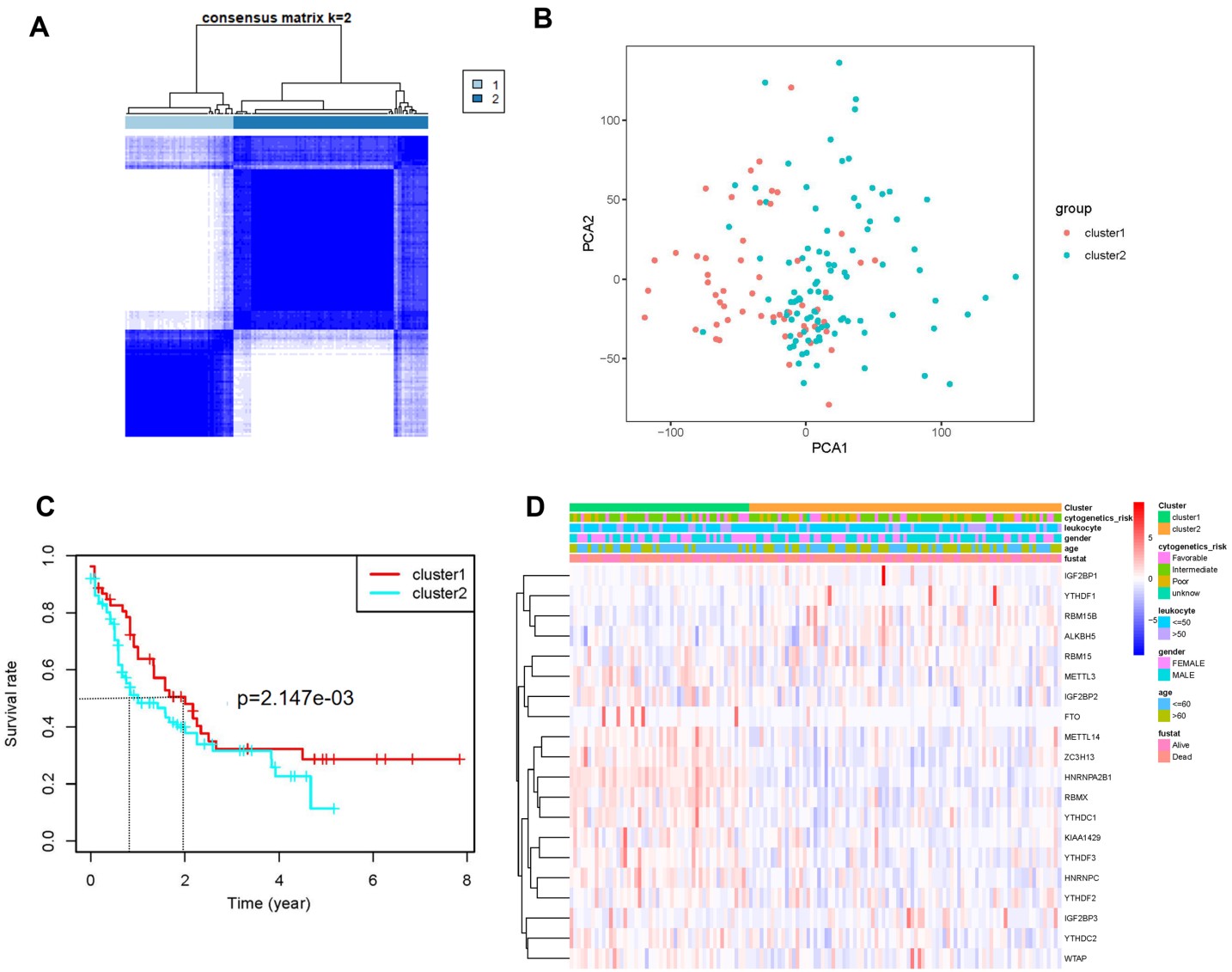

**Figure 2 Identification and analysis of two subgroups of 151 tumor samples demonstrating the distinct m6A expression.** (A) Consensus clustering matrix for $k = 2$. (B) Principal component analysis of the two subgroups. (C) Kaplan-Meier survival plots of the two subgroups. (D) Heatmap and clinicopathologic characteristics of the two clusters defined by the consensus expression of m6A RNA methylation regulators.

As *IGF2BP3* is highly expressed in patients with AML and is negatively correlated with patient survival ($P < 0.001$, Fig. 4B), compared to *HNRNPA2B1* ($P = 0.288$, Fig. 4C) and *YTHDF3* ($P = 0.032$, Fig. 4D). Thus, *IGF2BP3* was selected for further research. First, patients were classified using the median level of *IGF2BP3* expression as cut-off. Subsequently, chi-square test was used to perform a comprehensive analysis of the correlation between the level of *IGF2BP3* expression and clinical characteristics, such as age, gender, leukocyte count, and risk stratification (Table 2). The results revealed that the level of *IGF2BP3* expression was strongly associated with age and cytogenetic risk.

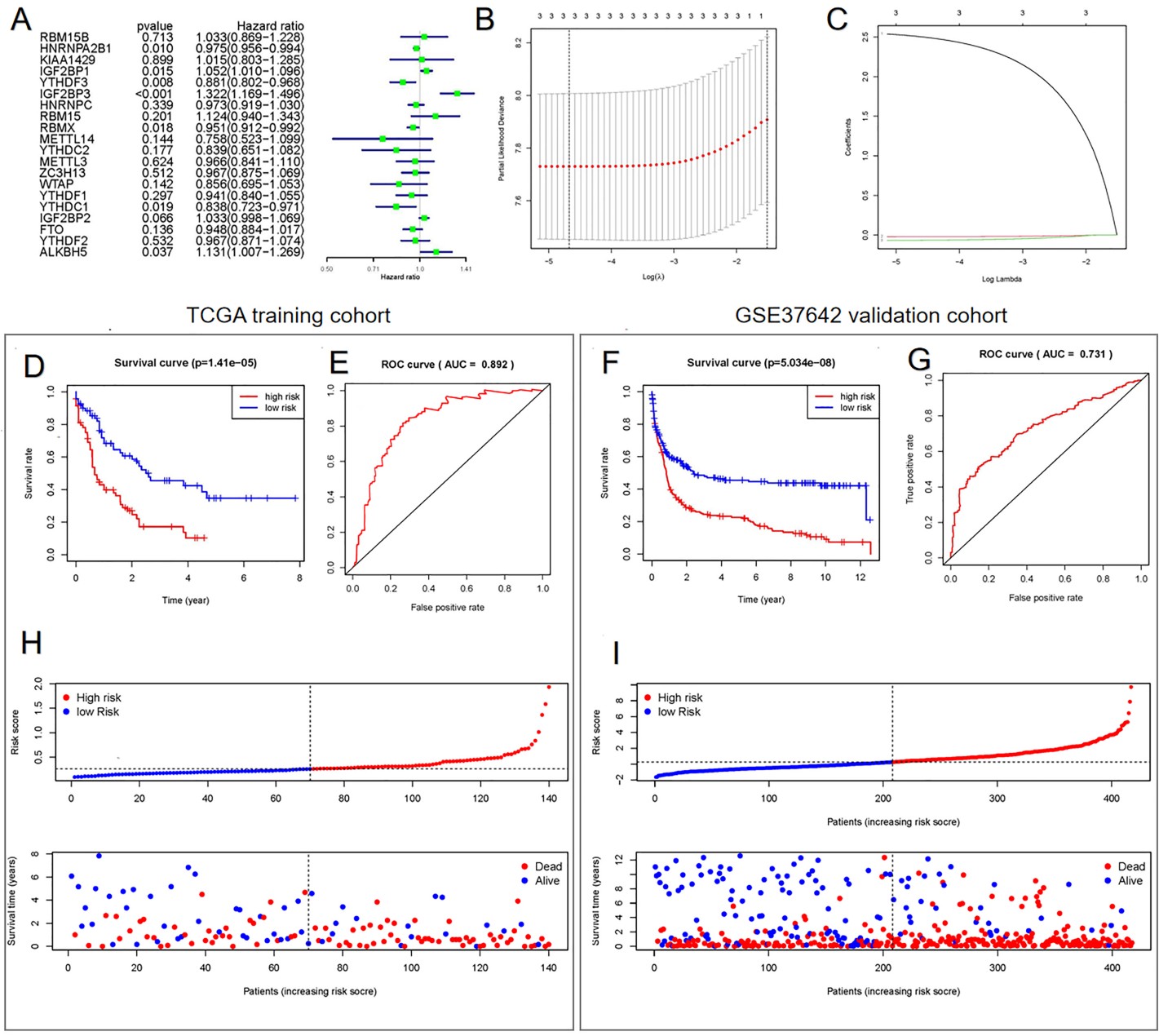

**Figure 3 Construction of the Cox regression model and validation of prognostic model in TCGA training cohort and GSE37642 validation cohort.** (A) Forest plots for hazard ratios of survival-associated m6A methylation-related genes in patients with AML. (B) Partial likelihood deviance *vs.* log (λ) was drawn using the LASSO Cox regression model. (C) Coefficients of selected features were indicated by lambda parameter. (D and F) Kaplan-Meier survival plots of the two cohort. (E and G) Receiver operating characteristic curves of the prognostic model in the two cohorts. (H and I) Risk score and survival status of each patient.

### Establishment of KG-1a and THP-1 cells with IGF2BP3 knockdown

We established cell lines stably expressing shIGF2BP3 to investigate the role of *IGF2BP3* in AML. The efficiency of shRNAs targeting *IGF2BP3* in KG-1a and THP-1 cells was validated at the protein (Fig. 5A) and mRNA (Figs. 5B and 5C) levels. Moreover, shIGF2BP3-2, and shIGF2BP3-3 were selected for subsequent experiments and renamed

**Table 1 Cox proportional hazard regression analysis for overall survival.**

| Variables | Univariate | | Multivariate | |
|---|---|---|---|---|
| | HR (95% CI) | *P* value | HR (95% CI) | *P* value |
| Age (≤60/>60 yr) | 1.041 [1.025–1.058] | 0.000 | 1.033 [1.016–1.050] | 0.000 |
| Gender (female/male) | 1.032 [0.671–1.586] | 0.887 | 0.683 [0.430–1.085] | 0.106 |
| Leukocyte | 1.433 [0.872–2.355] | 0.156 | 1.579 [0.946–2.636] | 0.081 |
| Cytogenetics risk | 1.842 [1.324–2.562] | 0.000 | 1.669 [1.169–2.383] | 0.004 |
| Risk score (high/low) | 5.732 [2.967–11.071] | 0.000 | 3.449 [1.629–7.304] | 0.001 |

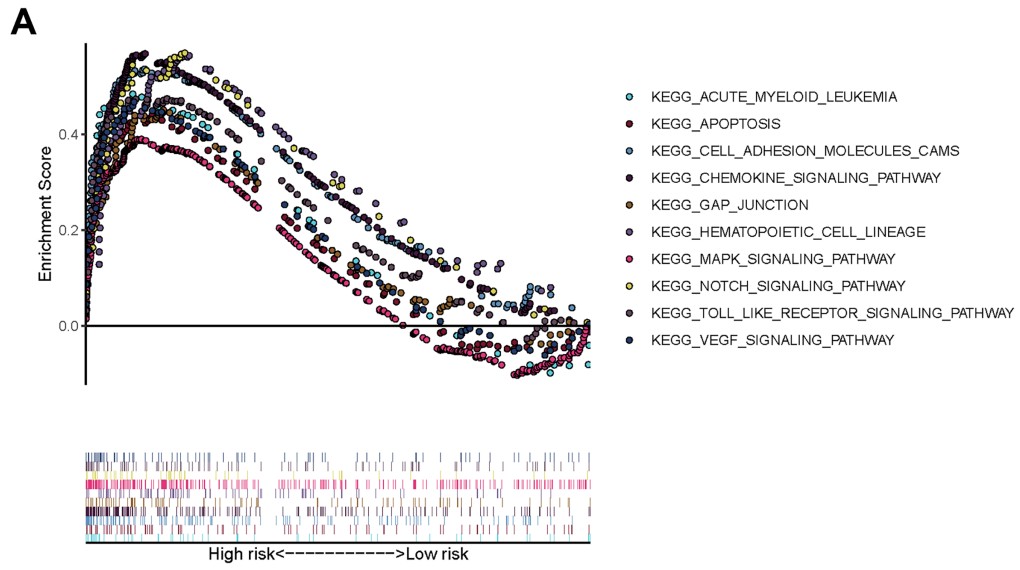

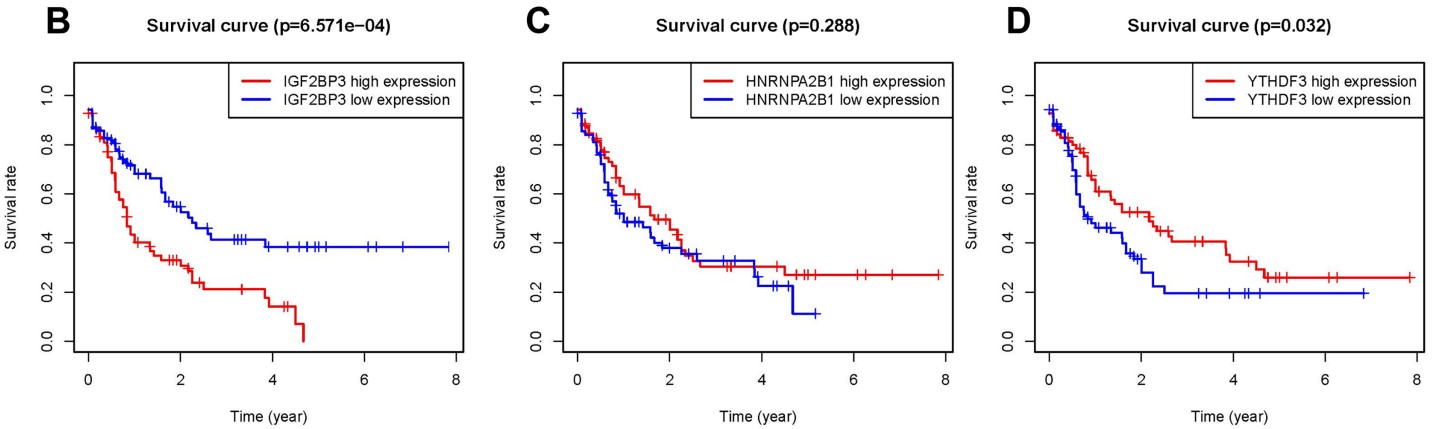

**Figure 4 Survival analysis and GSEA enrichment analysis.** (A) Enrichment of genes in the Kyoto Encyclopedia of Genes and Genomes (KEGG) in different pathways between the high- and low-risk groups using GSEA. (B and D) Kaplan-Meier curves of overall survival in patients with AML based on the three selected m6A RNA methylation regulators levels.

**Table 2  Relationship between IGF2BP3 expression and clinicopathological features in patients with AML.**

| Basic information | Total | IGF2BP3 expression | | P value |
|---|---|---|---|---|
| | | High (%) | Low (%) | |
| Age (yr) | 150 | | | |
| ≤60 | 83 | 35 (42.17) | 48 (57.83) | 0.033* |
| >60 | 67 | 40 (59.70) | 27 (40.30) | |
| Gender | | | | |
| Male | 83 | 39 (46.99) | 44 (53.01) | 0.412 |
| Female | 67 | 36 (53.73) | 31 (46.27) | |
| Leukocyte/white blood cell | | | | |
| $<50 \times 10^9$/L | 111 | 52 (46.85) | 59 (53.15) | 0.193 |
| $>50 \times 10^9$/L | 39 | 23 (58.97) | 16 (41.03) | |
| Cytogenetics risk | | | | |
| Favorable | 31 | 7 (22.58) | 24 (77.42) | 0.002** |
| Intermediate | 81 | 49 (60.49) | 32 (39.51) | |
| Poor | 36 | 18 (50.00) | 18 (50.00) | |
| Unknown | 2 | 1 | 1 | |

**Notes:**
* $P < 0.05$.
** $P < 0.01$.

as follows: KG-1a/THP-1-NC (*IGF2BP3* knocked down by shRNA), KG-1a/THP-1-SH1 (*IGF2BP3* knocked down by shIGF2BP3-2), and KG-1a/THP-1-11-SH2 (*IGF2BP3* knocked down by shIGF2BP3-3).

## IGF2BP3 promotes the proliferation of AML cells and the transition of cell cycle from G1 to S phases

The CCK-8 assay revealed that the knockdown of *IGF2BP3* had a significant effect on the viability of KG-1a and THP-1 cells (Figs. 5D and 5E). Moreover, we measured and analyzed the cell cycle distribution in KG-1a and THP-1 cells. Figure 5F show that the knockdown of *IGF2BP3* increased the ratio of G0/G1 phase cells but decreased the percentage of S phase cells. These findings suggested that *IGF2BP3* regulates the cell cycle by promoting the G1/S phase transition.

## IGF2BP3 inhibits the apoptosis and differentiation of AML cells

We used annexin V and PI stains to detect the effect of IGF2BP3 on the apoptosis of AML cells. Our findings suggested that *IGF2BP3* knockdown significantly promotes cell apoptosis (Figs. 5G and 5H). Additionally, the expression level of CD11b and CD14, which can be determined by qPCR, was used to identify the PMA-induced cell differentiation. mRNA expression of CD11b and CD14 was significantly higher in the AML group than in the negative control group (Figs. 5I and 5J). These results demonstrated that *IGF2BP3* knockdown had two effects on AML cells: induction of apoptosis and promotion of differentiation.

Peerj

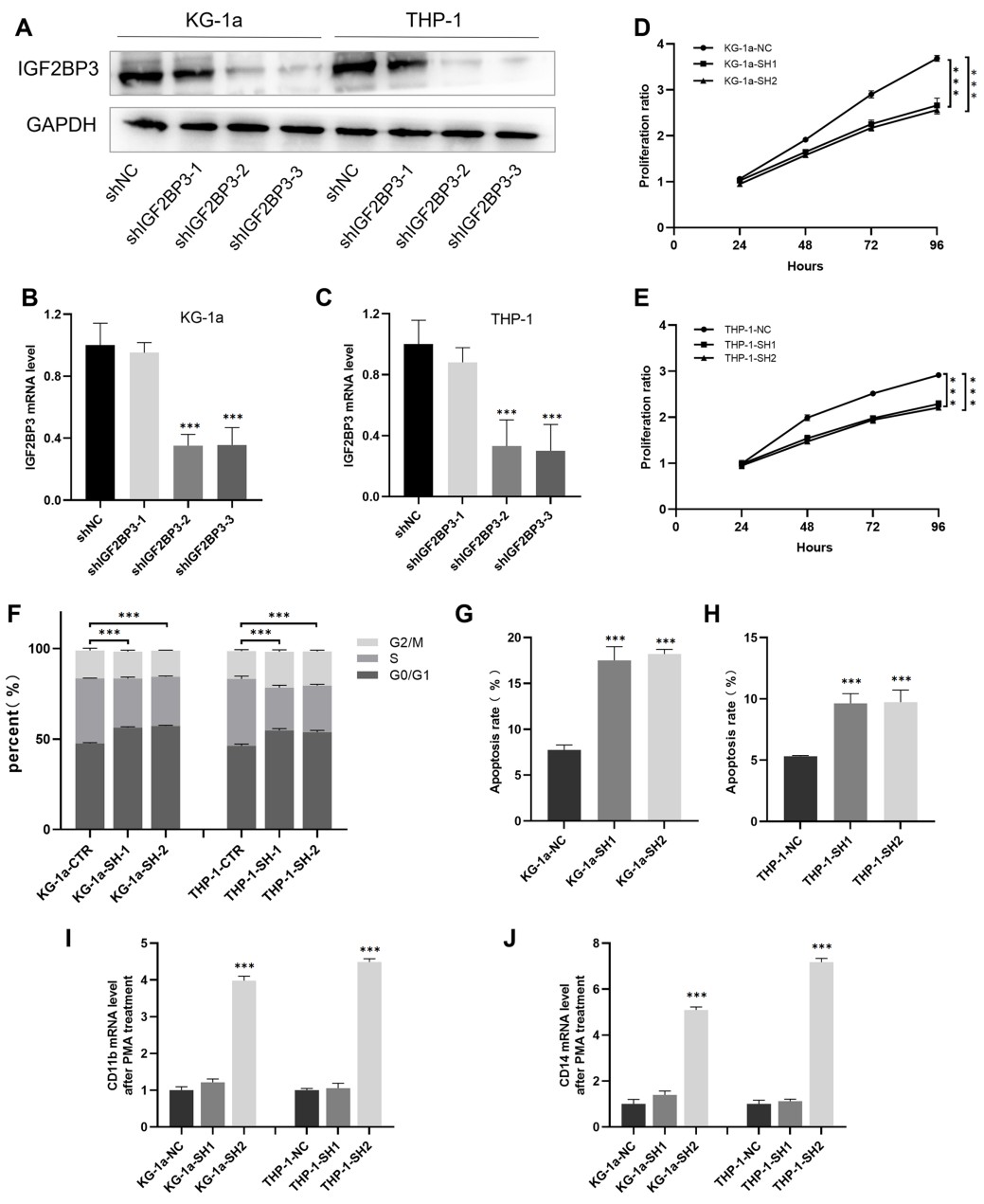

**Figure 5 Establishment of *IGF2BP3* knockdown cells and their biological phenotypes.** (A) IGF2BP3 protein expression levels in AML cells treated with shRNA1, shRNA2, shRNA3, or NC. (B and C) IGF2BP3 mRNA expression levels in AML cells treated with shRNA1, shRNA2, shRNA3, or NC. (D and E) Downregulation of *IGF2BP3* expression inhibited proliferation. (F) Downregulation of *IGF2BP3* induced G0/G1 phase arrest. (G and H) Downregulation of *IGF2BP3* increased apoptosis. (I and J) The mRNA level of CD14 and CD11b under PMA treatment. ***$P < 0.001$.

## IGF2BP3 promotes the proliferation of AML cells through the regulation of the JAK/STAT signaling pathway

To further explore the molecular mechanism of IGF2BP3 involved in AML progression, GSEA was conducted. The results revealed that *IGF2BP3* high-expression is positively

correlated with the JAK/STAT signaling pathway; notably, GSEA was conducted to investigate the possible mechanism by which *IGF2BP3* participates in the occurrence and progression of AML (Fig. 6A).

We focused on positive regulators upstream of the JAK/STAT pathway to investigate potential targets for IGF2BP3 in AML cells. Notably, the mRNA level of EPOR was clearly elevated in patients with AML (Fig. 6B). Furthermore, patients with high EPOR expression had a significantly shorter survival time than those with low EPOR expression ($P = 0.013$, Fig. 6C).

The activation of JAK/STAT signaling pathway was blocked because of the downregulation of *IGF2BP3*, as indicated by western blotting in Fig. 6D. Indeed, the expression of EPOR, phosphorylated JAK2, and phosphorylated STAT5 was decreased in *IGF2BP3* knockdown AML cells. Additionally, after IGF2BP3 depletion, c-Myc and bcl-2 —two important downstream targets of JAK/STAT signaling—were downregulated in KG-1a and THP-1 cells. The expression of EPOR in shIGF2BP3-transfected cells was determined by RT-qPCR (Figs. 6E and 6F). The findings suggested that silencing IGF2BP3 can reduce the expression of EPOR in AML cells at the mRNA and protein levels.

## IGF2BP3 affects the half-life of EPOR mRNA by regulating its m6A-methylated level

*IGF2BP3*, a newly discovered family of m6A readers, increases the stability of target mRNAs and extends the translation in a methylation-dependent manner (*Huang et al., 2018*). To investigate whether m6A mRNA methylation triggers IGF2BP3 to regulate EPOR expression, potential sites were predicted using an SRAMP tool based on sequence features (*Zhou et al., 2016*). The results indicated that EPOR mRNA has multiple predicted sites (Fig. 6G).

To verify whether m6A is involved in the regulation of EPOR by *IGF2BP3*, we added different concentrations of the m6A synthesis inhibitor cycloleucine to the culture medium for 48 h and monitored changes in EPOR levels. Notably, cycloleucine, a commonly used m6A inhibitor, is a competitive inhibitor of S-adenosylmethionine (SAM) transferase, and it provides methyl groups for the m6A process (*Chen et al., 2019*). Our findings indicated that the expression of EPOR was reduced in cycloleucine (m6A synthesis inhibitor)-treated cells in a concentration-dependent manner (Figs. 6H and 6I). The decrease in EPOR mRNA was more apparent after *IGF2BP3* knockdown. These results indicate that m6A affects the regulation of EPOR by *IGF2BP3*, and *IGF2BP3* binds to EPOR mRNA through m6A.

Methylated RNA immunoprecipitation (MeRIP) was performed to validate m6A methylation level of EPOR mRNA. The mRNA with m6A modification was recognized and bound with a specific m6A antibody; subsequently, the mRNA expression of EPOR was quantified by RT-PCR. The results suggested that *IGF2BP3* knockdown significantly decreased m6A levels of EPOR mRNA (Figs. 6J and 6K).

To further verify whether the m6A-methylated level had a critical effect on EPOR mRNA stability, we performed an assay for EPOR mRNA stability, and our results

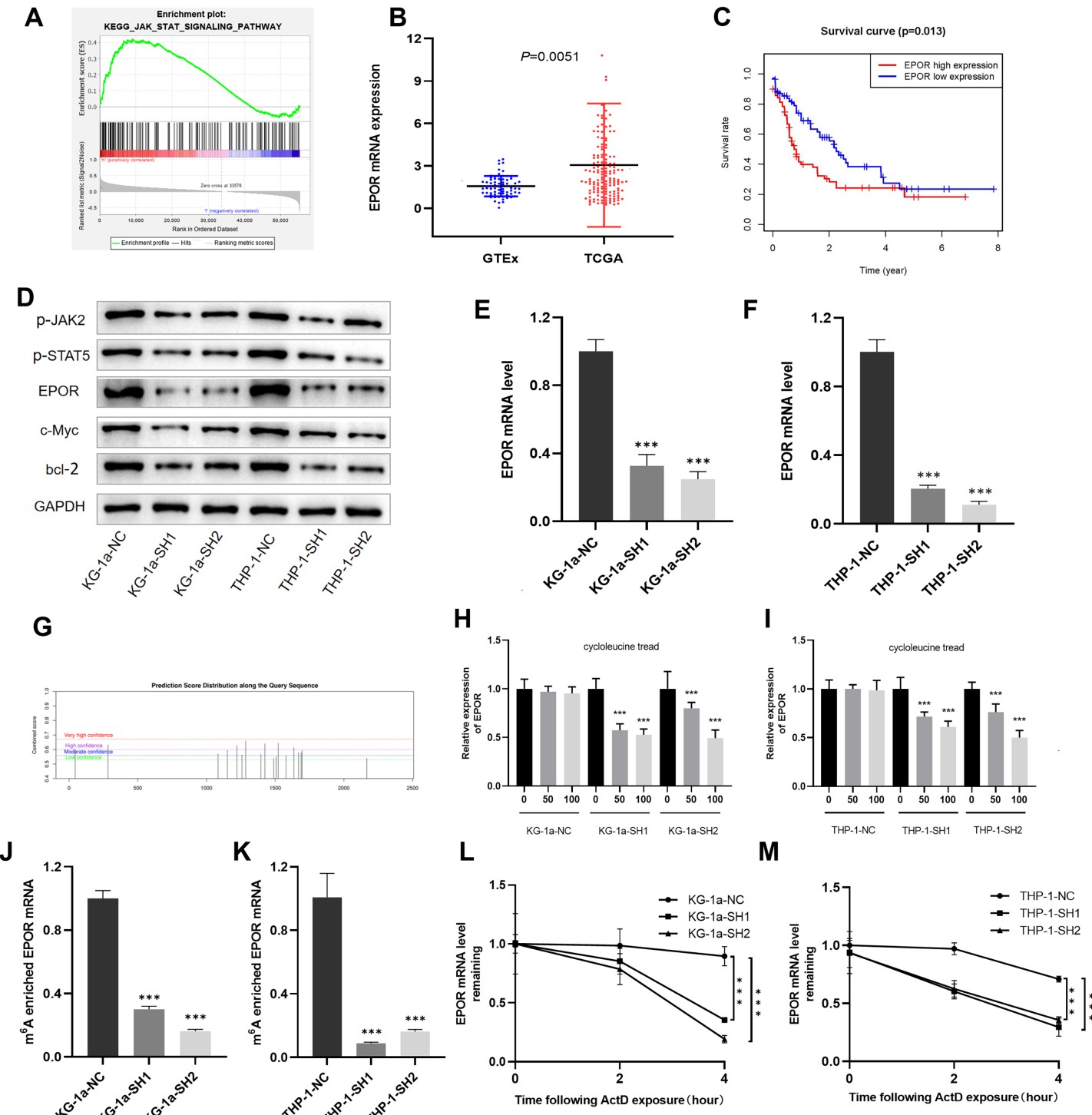

**Figure 6 EPOR was identified as a target of *IGF2BP3*-mediated m6A modification in AML.** (A) GSEA plot revealed that IGF2BP3 was positively associated with the JAK/STAT pathway. (B) EPOR was significantly higher in patients with AML than in normal individuals. (C) The high expression of EPOR suggested a poor prognosis. (D) Expression of the JAK/STAT pathway key genes was detected by western blotting. (E and F) The mRNA level of EPOR increased after knocking down *IGF2BP3*. (G) Predicted m6A site of EPOR mRNA. (H and I) Expression level of EPOR mRNA after treatment with different concentrations of cycloleucine. (J and K) The m6A methylation level of EPOR mRNA was reduced after knocking down *IGF2BP3*. (L and M) The stability of EPOR mRNA was decreased after knocking down *IGF2BP3*. ***$P < 0.001$.

confirmed that in the presence of IGF2BP, EPOR mRNA was more stable in AML cells (Figs. 6L and 6M).

Briefly, knockdown of *IGF2BP3* represses the m6A-methylated level of EPOR mRNA and inhibits the activity of JAK/STAT signaling pathway.

## DISCUSSION

AML is one of the most common cancers in adult patients. Despite advances in combination chemotherapy and bone marrow transplantation, AML, a highly heterogeneous disease, is associated with a high mortality rate. AML patients with various subtypes, karyotypes, gene expression patterns, and mutation types have varying prognoses. Therefore, it is important to investigate new prognostic biomarkers of AML and improve the diagnostic stratification and prognostic evaluation system. This can further optimize the appropriate treatment regimen and improve the survival rate of such patients.

Previous studies have examined the clinical relevance of 20 m6A regulators in 33 cancer types and reported that some of them are significantly correlated with cancer-associated pathways. Notably, *IGF2BP3* has been identified as a potential oncogene involved in the progression of various cancers. IGF2BP3 is an RNA binding protein that is primarily found in the cytoplasm. It regulates target RNA localization, stability, and translation as a meticulous post-transcriptional conditioner. Furthermore, it regulates tumorigenesis, cancer progression, tumor metastasis, and invasion, suggesting that *IGF2BP3* is a promising target for cancer therapies (*Li et al., 2019*). Previous studies have indicated that *IGF2BP3* plays an oncogenic role in colorectal (*Yang et al., 2020*), pancreatic (*Pasiliao et al., 2015*), lung (*Zhao et al., 2017*), and bladder (*Huang et al., 2020*) cancers. However, only a few studies have identified its role in AML carcinogenesis and as an m6A reader.

In the present study, we performed a more systematic analysis for determining the link between cancers and 20 m6A methylation genes, particularly with individual genes. We developed a three-gene model for prognostic analysis based on the expression of these genes and validated its efficacy. *IGF2BP3* was upregulated in AML based on public databases, and patients with higher *IGF2BP3* expression had a lower overall survival. Biological phenotypic studies revealed that the knockdown of *IGF2BP3* inhibited cell growth by preventing cell cycle progression from G1 to S phase and induced cell apoptosis and differentiation. These results clearly indicate that *IGF2BP3* is related to AML progression. Our study focused on the potential mechanism of *IGF2BP3* in AML and identified its effects on the JAK/STAT signaling pathway *via* m6A modification of EPOR.

EPOR is an erythropoietin (EPO)-specific transmembrane receptor. EPO and EPOR were found in 24 different types of malignant tumors, including leukemia (*Yasuda et al., 2003*). Notably, as a process upstream of the JAK/STAT pathway, EPO induces JAK2 phosphorylation and activates STAT5 by binding with EPOR, and its dysfunction has critical effects on various activities, such as cell growth, cell apoptosis, and angiogenesis (*Annese et al., 2019*). In the present study, we identified EPOR mRNA as a new downstream target of *IGF2BP3*, and the suppression of *IGF2BP3*-mediated cell growth is dependent on destabilization of the EPOR mRNA in AML cells. EPOR is a major upstream

target of the JAK/STAT pathway, explaining why IGF2BP3 is closely related to AML progression.

Previous studies have reported that *IGF2BP3* plays oncogenic roles by activating PI3K/AKT pathway in colorectal cancer (*Zhang et al., 2020*), Ewing sarcoma (*Mancarella et al., 2018*), and glioblastoma (*Suvasini et al., 2011*). *IGF2BP3* is highly expressed in bladder cancer and may sustain cell growth by activating the JAK/STAT pathway (*Huang et al., 2020*). The JAK/STAT signaling pathway has various biological effects, including cell growth or amplification, differentiation, inflammation, and apoptosis. The abnormal aberrant activation of the JAK/STAT signaling pathway is associated with the development of AML (*Venugopal, Bar-Natan & Mascarenhas, 2020*), and JAK2 inhibitors can reverse this effect (*Habbel et al., 2020*; *Moser et al., 2021*). Moreover, STAT5 is a potential target for patients with refractory and relapsed AML (*Brachet-Botineau et al., 2020*; *Page et al., 2012*). Therefore, inhibition of JAK2 and STAT5 may be a targeted therapy for AML.

## CONCLUSIONS

Our study reported that *IGF2BP3* is an oncogenic gene that is overexpressed in AML. Furthermore, IGF2BP3 promotes AML progression by regulating the m6A level and mRNA stability of EPOR, which further influences the activity of the JAK/STAT signaling pathway. These findings indicated that the IGF2BP3-m6A-EPOR axis is important in AML, implying that it can predict patient survival and that targeting this axis may be an effective strategy for AML therapy. However, *in vivo* studies are warranted to provide direct genetic evidence of IGF2BP3 in AML pathogenesis. The signal transduction pathways activated by EPO/EPOR compound also include PI3K/AKT and RAS/MAPK, indicating that the JAK/STAT pathway is not the only pathway involved in IGF2BP3/EPOR regulation.

### Funding
The publication costs were funded by grants from the Major Research and Development Plan of Shandong Province (2021LCZX05) and the ECCM Program of Clinical Research Center of Shandong University (No. 2021SDUCRCB009). The funders had no role in study design, data collection and analysis, decision to publish, or preparation of the manuscript.

### Grant Disclosures
The following grant information was disclosed by the authors:
The Major Research and Development Plan of Shandong Province: 2021LCZX05.
The ECCM Program of Clinical Research Center of Shandong University: 2021SDUCRCB009.

## Competing Interests

The authors declare that they have no competing interests.

## Author Contributions

- Jin Fan conceived and designed the experiments, authored or reviewed drafts of the article, and approved the final draft.
- Mengqi Zhuang performed the experiments, analyzed the data, prepared figures and/or tables, and approved the final draft.
- Wei Fan performed the experiments, analyzed the data, prepared figures and/or tables, and approved the final draft.
- Ming Hou conceived and designed the experiments, authored or reviewed drafts of the article, and approved the final draft.

## Data Availability

The raw data and uncropped gels/blots are available in the Supplemental Files.

## Supplemental Information

Supplemental information for this article can be found online at http://dx.doi.org/10.7717/peerj.15706#supplemental-information.

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
