# Peer review of "RNA N6-methyladenosine reader IGF2BP3 promotes acute myeloid leukemia progression by controlling stabilization of EPOR mRNA"

_PeerJ, doi:10.7717/peerj.15706_

## Round 0.1 · original submission · Major Revisions

Authors should revise according to the suggestions of reviewers. The modifications should be marked. A point-to-point response letter is needed.

·

Basic reporting

- Please, add units for Figure 1B. Also, there is a minor correction in the legend (green->blue).

- There is a bit difficult to visualize r-coefficients in Figure 1C.

Experimental design

- Please add the supplier and product ID of all reagents (for example, RIPA, line 106, vector line 98).

- Use conventional units for all methodological steps. For example, line 107 (1200 r/min -> RCF)

- Please, add information about primers used for the housekeeping region in qPCR experiments.

Validity of the findings

- Please, add details of the methods for elaborating Figures 1, 2, and 6A. There is some missed information. For example, how many samples from TCGA were included? Was that data downloaded directly from the source (TCGA)? If so, was it processed?

- According to Figures 5B-C, shIGF2BP3-1 affects the RNA level expression of the gene, at a similar level to shIGF2BP3-2 and shIGF2BP3-3. However, it is inconsistent with Figure 5A which shows that shIGF2BP3-1 has no effect in reducing IGF2BP3 protein levels. Could you discuss it, please?

- Also, if shIGF2BP3-1 and shIGF2BP3-2 showed different level of IGF2BP3, how can they produce similar results at Figures 5D,E,G, and H? Would it still be caused by IGF2BP3?

- I believe that the statement of lines 200-201 deserves revision. "The analysis of overall survival revealed that the survival time of subgroup 2 was higher than that of Subgroup 1 (Figure 2C)". The p-value was not statistically significant.

- About Figure 3D, did you use a group of samples to build a model and then test this formula on the same cohort? If so, I think it's a bit redundant. It would be interesting to use a training subcohort to calculate the model and then test in a validation subcohort.

·

Basic reporting

'no comment'

Experimental design

In this interesting article the authors demonstrated the relevance of m6A RNA methylation for the performance of AML cells. However, some aspects need to be reviewed:

1. As first analysis, the authors compared the expression of 20 genes related to m6A RNA methylation between AML samples and controls. Then, downregulated, and upregulated genes were identified. However, the control group correspond to peripheral blood (PB) from healthy individuals (GTEx), while the AML samples correspond to bone marrow (BM) samples. PB and BM are different tissues. PB are main composed by mature cells, while BM by immature hematopoietic cells plus other microenvironment components like fibroblasts. For that, when this type of comparison is made, the control cells are BM CD34+ cells. Also, because the different hierarchies between AML blasts, other progenitors like CMPs and GMPs are usually included. PB is a tissue lacking progenitors, therefore, there is no biological justification to use PB as control for AML. There are several publicly available databases that include healthy progenitors and AML samples to perform that analysis, like BloodSpot (https://servers.binf.ku.dk/bloodspot/). Finally, Do the authors made a normalization process to merge and subsequently compare TCGA data with GTEx data? There is nothing in the methodology about that.

2. The authors affirm in the line 200 of the manuscript that “The analysis of overall survival revealed that the survival time of subgroup 2 was higher than that of subgroup 1”. Sorry but the KM plot is not showing that. Also, the median survival is a good parameter to compare groups in survival analysis, and subgroup 2 has a shorter median survival than subgroup 1. Finally, the p > 0.05 of the KM indicates that there a no difference in the survival between these two subgroups.

3. Possible there is something wrong with the ROC curve showed in the Figure 3E. Review carefully the process what this curve was made.

4. In the line 227, the authors said “strongly associated” when referred to the relation between the score with age and cytogenetic risk. However, the multivariate regression indicated that the variance given by the score to the overall survival is not totally explained for these variable. Take care at the time to describe this apparent conflicting results.

5. The authors affirm that YTHDF3 does not have relation with survival, but at the same the showed a log rank p-value <0.05.

6. In Figure 6B the authors indicates IGF2BP3 mRNA expression, but in the text mentioned that the expression of EPOR gene is showed. Additionally, again, this comparison was made between PB and BM AML samples. The Figure 6D that showed less EPOR protein abundance when IGF2BP3 is knocked down is enough and more reliable method to affirm relation between IGF2BP3 and EPOR in AML cells.

Validity of the findings

The key point of this work is the relation between IGF2BP3 and the performance of AML cells, possible by regulating EPOR. However, if the authors want to convince about the clinical applicability of their score, them, other AML cohorts need to be evualuated. There are several publicly available AML datasets other than TCGA, like BeatAML, HOVON, TARGET, AMLCG 1999, AMLCG 2008, and so on.

Additional comments

'no comment'

---

## Round 0.2 · Minor Revisions

Authors should revise according to the suggestions of reviewers. The modifications should be marked. A point to point response letter is needed.

·

Basic reporting

no comment

Experimental design

no comment

Validity of the findings

no comment

·

Basic reporting

no comment

Experimental design

Minor revisions:
In the rebuttal letter the authors indicated that the TCGA samples of the AML patients correspond to peripheral blood (PB) and not bone marrow (BM), assuming that there is no problem with the comparison between AML samples and normal peripheral blood from GTEx. However, with this answer the author are ignoring the biological reason for what is not adequate this type of comparison. Firstly, AML cells, independently if collected from PB or BM, are cells that are in a “differentiation blocking state”, and there is nothing similar in PB, but there is in the bone marrow. Secondly, AML cells have hierarchies, with some AMLs with higher percentage of CD34+ cells (totally absent in PB), others with immunophenotypes with more progenitors-like cells, like GMPs (also absent in PB), and other with more mature phenotypes, like monocytic leukemias. Therefore, the comparison of AML cells with normal cells needs to be adjusted to a comparison with a more similar counterpart normal tissue. Thus, some authors directly compare the AML cells with healthy CD34+ cells, which is assuming that all AML cells have enough CD34+ cells to get their transcriptome in the RNA-seq analysis of the bulk sample. Other alternative is to compare AML cells versus total healthy bone marrow. Finally, other authors consider the complexity of the AML samples and compare the AML cells with normal hematopoietic cells in different states of differentiation. All this possibilities have biological sense. In conclusion, the authors need to rethink the methodology of this part of their work, or complement their results with more adequate comparison.

The authors indicated that it is all ok with their AUC curve. When an AUC curve is performed, the sensitivity and the 1-specificity values are organized in a table from 0 to 1. However, in the AUC figure (Figure 3E) presented for the authors, the red line sometimes goes backwards in the x and y axes, which are not typical in a AUC curve.

In the Figure 1B, in the name of the Y axes said "FRPM" insteat of FPKM. Also, please include legend for this figure.

Validity of the findings

no comment

Additional comments

no comment

---

## Round 0.3 · accepted · Accept

The authors have addressed the reviewers' concerns properly and revised the manuscript accordingly. The manuscript can be accepted for publication in its current form.

·

Basic reporting

No comment

Experimental design

In this third review, the authors mantained the comparison between AML samples and peripheral blood samples, arguing some relevant literature. Despite the authors modified the methodology, the ideia of this new comparison is the same with the previous version. Therefore, the authors were unable to resolve a simple observation. There are several public datasets with transcriptomes of healthy bone marrow, progenitors or hematopoietic stem cells, which are better control samples for AML.

Considering that in this work, this comparison (AML vs control) is not critical for the others analysis, I will not made additional comments.

Validity of the findings

No comment

Additional comments

No comment